# Plastic Waste Upcycling: A Sustainable Solution for Waste Management, Product Development, and Circular Economy

**DOI:** 10.3390/polym14224788

**Published:** 2022-11-08

**Authors:** Rajkamal Balu, Naba Kumar Dutta, Namita Roy Choudhury

**Affiliations:** 1Chemical and Environmental Engineering, School of Engineering, RMIT University, Melbourne, VIC 3000, Australia; 2ARC Industrial Transformation Research Hub for Transformation of Reclaimed Waste into Engineered Materials and Solutions for a Circular Economy (TREMS), RMIT University, Melbourne, VIC 3000, Australia

**Keywords:** plastic waste, upcycling, vitrimers, nanocomposites, 3D printing, aromatic products, biopolymers, sustainability, circular economy

## Abstract

Plastic waste pollution, including non-biodegradable landfills, leaching of toxic chemicals into soil and waterways, and emission of toxic gases into the atmosphere, is significantly affecting our environment. Conventional plastic waste recycling approaches generally produce lower value materials compared to the original plastic or recover inefficient heat energy. Lately, upcycling or the valorization approach has emerged as a sustainable solution to transform plastic waste into value-added products. In this review, we present an overview of recent advancements in plastic waste upcycling, such as vitrimerization, nanocomposite fabrication, additive manufacturing, catalytic transformation, and industrial biotechnology, envisaged with technical challenges, future developments, and new circular economy opportunities.

## 1. Introduction

Plastics are synthetic or semi-synthetic materials made of polymers as a main ingredient. In the 21st century, plastic has become an integral part of our daily life. It is used in almost every sector, such as packaging, consumer products, medical devices, textiles, transportation, building and construction, electrical and electronics industries, etc., owing to its low density, high strength-to-weight ratio, durability, high chemical resistance, thermal and electrical insulation, mouldability, and low production cost [1]. Almost all the plastics currently used today are produced from non-renewable fossil fuels (petroleum byproducts) with a high carbon footprint (i.e., the total amount of greenhouse gases (GHG), including carbon dioxide (CO_2_) and methane that are generated in its life cycle) [2]. Some of the commonly used household plastic materials include polyethylene terephthalate (PET), low-density polyethylene (LDPE), high-density polyethylene (HDPE), polyvinyl chloride (PVC), polypropylene (PP), polyurethane (PU) and polystyrene (PS). These plastics are generally used once and then disposed as landfills, recycled (denoted by the letter ‘r’ before abbreviation), or incinerated due to their non-biodegradable nature [2]. In 2015, the global life cycle GHG emissions of above plastics (excluding any carbon credits from recycling) were estimated as 1.8 gigaton (Gt) of CO_2_-equivalent (Figure 1a), with the resin-production stage generating the highest emissions (61%), followed by the conversion stage (30%) and end-of-life (EoL) stage (9%) [3]. We generated about 7500 million metric tons (MMt) of plastic waste by the year 2020, which is projected to reach 26,000 MMt by the year 2050 (Figure 1b) [4]. Today, only about 9% of the plastic waste is recycled and 12% is incinerated, while the remaining 79% has accumulated in landfills, rivers, and oceans, which impacts our environment and ecosystem [5]. To combat the escalating plastic waste problem and achieve economic growth, environmental protection, and societal benefits, a circular economy model with the 6R principles; including reduce, recover, reuse, recycle, redesign, and remanufacture, has been realized lately in many countries [6].

The plastic waste products commonly dumped in soil and waterways leach out toxic chemicals, such as phthalates and bisphenol A, into the surrounding environment, which pose serious health risks to soil, microorganisms, vegetation, marine life, and animals [7]. Conversely, incineration of plastic waste liberates halogens, dioxins, furans, and other hazardous substances, which cause serious damage to the environment and ecosystem [8]. Therefore, as a sustainable approach towards a cleaner and healthier environment, recycling and upcycling technologies have been developed to minimize accumulation and achieve valorization of plastic waste. Recycling is the reprocessing of waste into new and useful products. All plastic recyclables fall into three main types of recycling: primary, secondary, and tertiary [9]. Primary recycling refers to recovery and reuse (usually for the very same purpose) of plastic without altering its current state. Secondary recycling (or mechanical recycling) refers to reprocessing of plastics by physical means, such as shredding, granulation, and extrusion, whereas tertiary recycling (chemical or feedstock recycling) refers to decomposition of plastics to monomers or other valuable low molecular weight fragments [9], which has also led to the development of degradable plastics [10]. Comprehensive details of various recycling technologies can be found elsewhere [11]. However, traditional approaches of recycling plastic waste via the first two methods produce lower quality materials, in terms of thermal, optical, barrier, and mechanical properties, because of the repeated degradation of their polymeric chains with each recycling. On the other hand, specific chemical recycling is not efficient enough to be used for each polymer nor is it economically sustainable [12].

Over the last decade, upcycling or creative reuse has emerged as a promising alternative to recycling plastic waste. In the upcycling approach, the plastic waste is reused in such a way as to create a product of higher quality or value than the original, which provides a technical solution for smarter waste management, new product development, and the circular economy. State-of-the-art upcycling concepts (compounding, reforming, transformation, etc.) involve physical (extrusion, electrospinning, printing, etc.), chemical (pyrolysis, gasification, hydrogenolysis, etc.), and biological (enzyme treatment, biosynthesis, etc.) methods to convert plastic waste into polymers (depolymerization–repolymerization, functionalization, etc.), molecules (monomers, fine chemicals, additives, etc.), and materials (carbon-based nanomaterial, blend compatibilizers, etc.) [13]. However, plastic waste upcycling is still in its infancy (Figure 1c). In this review, recent developments (the last four years) in polymer waste upcycling approaches, including vitrimerization, nanocomposite fabrication, additive manufacturing, catalytic transformation, and industrial biotechnology for producing various value-added products are summarized, and their applications, technical challenges, and future directions are discussed. Plastic waste upcycling generally involves sorting or separation of specific plastic waste, followed by decontamination prior to mechanical, thermal, chemical and/or biotechnological processing.

## 2. Upcycling of Plastic Waste

### 2.1. Vitrimerization

In recent years, vitrimerization has emerged as a promising novel approach to reprocess and recycle intractable waste via dynamic chemistry, which involves dynamic covalent bonds, a special type of covalent bond. Dynamic bonds can be dissociative or associative under external stimuli. Materials synthesized with dynamic covalent bond crosslinks are commonly referred to as covalent adaptive networks (CANs) [14]. Vitrimerization is the process of creating ‘vitrimers’, a new class of plastic materials with associative dynamic covalent bond crosslinks, where the network integrity is maintained during bond exchanges, whereas the network topology is constantly rearranged. Therefore, vitrimers combine the property advantages of thermoplastic and thermoset materials, such as re-processability, healability, recyclability, shape-memory behavior, and self-adhesion [15]. The vitrimer concept developed for commercial plastic materials can be potentially applied for their recycled waste, including polyesters, such as PET (or any thermoplastics containing ester bonds can be upgraded to vitrimers), and polyolefins (HDPE and PP) [16]. Vitrimer based on commercial PET has been developed by incorporating polyol (containing a tertiary amine structure) into the chain of PET (to furnish reactive hydroxyl groups) and reacting it with diepoxy to obtain the dynamic crosslinked networks [17]. The obtained vitrimer exhibited improved thermal and mechanical properties compared to neat PET, and demonstrated excellent re-processability via extrusion, compression, and injection molding suitable for large-scale industrial production. Caffy et al. [18] synthesized vitrimer from commercial HDPE via a single-step reactive extrusion by combining nitroxide chemistry (for radical grafting of 2,2,6,6-Tetramethyl-4-((2-phenyl-1,3,2-dioxaborolan-4-yl)-methoxy)piperidin-1-oxyl (TEMPO-BE) onto polyethylene) and boronic ester metathesis (as an associative exchange reaction). On the other hand, Saed et al. [19] developed a new extrudable vitrimer from PP, which was functionalized with maleic anhydride (MA) and dynamically crosslinked through thiol–thioester bond exchange using a transesterification catalyst 1,5,7-triazabicyclo[4.4.0]dec-5-ene. The PP vitrimer was demonstrated to be readily re-processable (recycled, remolded, rewelded, and 3D printed) multiple times, and exhibited 25% higher mechanical strength compared to the original PP, with a maximum gel fraction reaching about 55%.

Recently, Kar et al. [20] demonstrated the upcycling of PP bottle waste (PP_b_) and PE packaging waste (PE_p_) into re-processable high-performance vitrimers (with a gel fraction of 58% and 66%, respectively) using melt extrusion processing. The vitrimers were synthesized by first grafting the plastics with MA, followed by crosslinking with bisphenol A diglycidyl ether (DGEBA) using zinc acetylacetonate hydrate (Zn(acac)_2_) as the transesterification catalyst (Figure 2a). The vitrimers exhibited thermo-reversible associative bond exchange (Figure 2b), thermally triggered shape-memory behavior (Figure 2c) (with 90% recovery after multiple cycles), and superior mechanical stability compared to the original materials (Figure 2d) [20]. Moreover, development of vitrimer-based composite materials has also been realized, where an increase in filler concentration generally increases vitrimer temperature, mechanical properties, and self-healing properties [21]. These newly developed vitrimer systems have potential applications in a wide range of industrial sectors, including automotive, aerospace, electronics, and biomedical fields [22].

### 2.2. Nanocomposite Fabrication

In nanocomposite fabrication, advanced functional materials with tailored properties are developed by incorporation of functional nanofillers into the plastic waste matrix at desired concentrations. The cost, quality, and application of these nanocomposites depend on the type of plastic waste, property, and quantity of the incorporated nanomaterial, as well as the processing route used for composite fabrication. The fabricated composites can be used as such (this section) or can be thermochemically transformed into carbonaceous composites (see Section 2.4). A summary of value-added nanocomposite structures fabricated using plastic waste is provided in Table 1. Waste plastic-based composites fabricated using waste wood, rubber, and crushed glass are generally considered recycled composite materials rather than upcycled materials. Nanocomposites can be fabricated by thermal, mechanical, and solution processing methods.

Yasin et al. [23] reported a facile strategy for fabricating PET waste into PET nanofibrous membrane embedded with copper oxide (CuO) nanoparticles (NPs) by electrospinning, where the CuO NPs were synthesized using plant extract and mixed with PET waste solution (Figure 3a). The authors demonstrated the photocatalytic efficiency of the fabricated nanocomposite membrane for removal (99% efficiency) of methylene blue (MB) dye, which has potential applications in water treatment and filtration. CuO NPs prepared by the combustion method have also been used in the fabrication (melt mixing and compression molding) of nanocomposite sheets with HDPE waste. The nanocomposite sheets exhibited increased electron density, mass attenuation coefficient, and effective atomic number for γ-ray energies, which have potential applications in enhanced radiation-shielding [24]. In a separate study, Fan et al. [25] demonstrated the application of PE film waste-based porous nanocomposite membranes for UV shielding. The nanocomposites were fabricated by mixing PE waste granules with silicon dioxide (SiO_2_) and titanium dioxide (TiO_2_) NPs in liquid paraffin, followed by extrusion, granulation, and thermal compression molding. The fabricated composite membrane exhibited tensile strength of 8.4 MPa and a UV protection factor of 1500+ [25]. Wang and co-workers [26] reported a facile route to produce nanocomposites for electronic packaging applications using aluminum (Al)-plastic package waste (APPW). The APPW comprising 70% LDPE, 15% Al, and 15% PET was first finely powdered and mixed with graphite nanoplatelets (GNPs) using solid-state shear milling (S3M) technology (Figure 3b), followed by extrusion and injection molding to obtain a high thermally conductive (1.7 W/mK) and high electrically insulating (conductivity of 10^−10^ S/cm) nanocomposite. This work was further extended to fabricate nanocomposites using multilayer plastic package waste comprising 80% LDPE and 20% polyamide, where surface-oxidized Al nanoflakes were powder mixed at different ratios and compression molded into sheets exhibiting thermal conductivity in the range of 1.4–4.8 W/mK and high electrical insulation (conductivity of 10^−13^ S/cm) [27].

Conversely, Assis et al. [28] fabricated PS foams impregnated with tin oxide (SnO_2_) NPs using a thermally induced phase separation (TIPS) method, which exhibited a photodegradation efficiency of 98.2% for rhodamine B dye under UV irradiation and can be potentially applied for water treatment and filtration applications. Recently, Uddin et al. [29] fabricated superhydrophobic nanocomposites fibers using recycled expanded PS. The nanocomposite membranes were electrospun from PS solutions comprising various proportions of TiO_2_ NPs and Al microparticles, which exhibited a water contact angle of 157°. The fabricated membranes were demonstrated for fog-harvesting capability with daily water productivity of >1.35 L/m^2^. In addition, plastic waste-based transformed nanocomposites, where the plastic is converted into carbonaceous material in the final composite product, have also been reported. For example, Mir et al. [30] reported the synthesis of molybdenum carbide carbon (Mo_2_C) nanocomposites using plastic waste (pipette tips) and molybdenum trioxide via an in-situ carburization route. The obtained Mo_2_C nanocomposite has potential for hydrogen production and energy storage applications. However, such transformations are considered as chemical upcycling rather than nanocomposite formulations.

### 2.3. Additive Manufacturing

Additive manufacturing, or three-dimensional (3D) printing, is a constructive technique for building 3D objects from digital models. The 3D printing of plastics has gained increasing research attention in recent years due to its remarkable potential for fabricating complex structures, customizing the product at will, and reduced lead time and waste, which are advantages in comparison to many traditional manufacturing processes commonly used in industries [31]. Fused filament fabrication or fused deposition modeling (FDM) is the most widely used extrusion-based 3D printing technology for fabrication of value-added products from common polymer-based waste materials [32]. In FDM 3D printing, thermoplastic filaments are heated to their melting point in a nozzle head and deposited as polymer melt in a layer-by-layer fashion on a temperature-controlled bed. Lately, a low-cost, closed-loop, and low-carbon-footprint recycling approach has been realized for the circular economy by utilizing used thermoplastics as feedstock material for fabrication of 3D printing filament using milling and screw extrusion techniques [33,34]. However, the quality of 3D-printed plastic material, such as crystallinity, morphology, thermal, rheological, and mechanical properties, decreases with successive grinding and extrusion events [35]. Therefore, to account for such changes and to improve the property, quality, and value of printed structures, a variety of approaches, such as the addition of additives to control crystallinity, micro-nano fillers or reinforcing agents to improve mechanical/electrical property, rheology modifiers to improve printability, and blending of recycled plastic with virgin material or with another polymer/polysaccharide, have been explored [36]. These 3D printable blends or composite systems not only have the potential to overcome the processability and property limitations of pristine polymer systems, but also provide an opportunity to manufacture customized complex 3D engineering structures and industrial products on demand. A summary of value-added structures that are 3D printed by FDM using blend or composite filaments made from plastic waste is provided in Table 2. While the manufacturing technology of composites remains the same when natural polysaccharides and synthetic polymers are added, the processing conditions are tuned to suit their physical properties.

Incorporation of biochar has been recognized to improve mechanical, thermal, and electrical properties of polymer composites [47]. Idrees et al. [37] reported the fabrication of melt-compounded recycled PET (rPET)/biochar composite filaments by single-screw extrusion at 250 °C, where the biochar was derived from pyrolysis of packaging waste. The 3D-printed structures from a 5 wt% biochar composite filament (Figure 4a) showed about 60% increase in tensile modulus over neat PET. Carrete et al. [38] fabricated melt-compounded rPET/cellulose fiber composite filaments by twin-screw extrusion, where the cellulose fibers were derived from denim textile waste. The composite filaments were then 3D printed, where a 10 vol% loading of cellulose fibers showed a 62% increase in impact resistance and 64% increase in impact strength over neat PET. Conversely, Bex et al. [39] reported the 3D printing of rPET/continuous carbon filament fibers (CCFs) composite using a co-extrusion-type fused filament fabrication (FFF) printer, where a 25 wt% loading of CFFs showed more than a 10-fold increase in tensile strength over neat PET. For semicrystalline plastics like HDPE, crystallization-induced shrinkage (warpage) is a problem during 3D printing. To overcome this issue, Gudadhe et al. [48] compounded waste-derived HDPE with 10% LLDPE and 0.4% dimethyl dibenzylidene sorbitol (DMDBS) using a twin-screw extruder at 190 °C. The extruded blend filaments were 3D printed at 230 °C, which showed a significant decrease in warpage (<0.6 mm for the 10 mm tall bar). Mejia et al. [40] compounded waste-derived HDPE (90%) with 5% PP and 5.0% PP grafted with maleic anhydride (PP-MAh) using a single-screw extruder at 160 °C. The blend filaments were then 3D printed, which exhibited a 39% increase in tensile yield stress, and a 2.7-fold decrease in strain over neat HDPE. Conversely, Borkar et al. [41] reported the fabrication of melt-compounded rHDPE/carbon fibers (CF) composite filaments by twin-screw extrusion at 200 °C, where the CF was derived from dry offcut fabric (Toray T300 grade). The 3D-printed structures from a 29.5 vol% CF composite filament showed about 11% increase in tensile yield and 188% increase in tensile strength over neat HDPE. Zander et al. [42] reported the fabrication of rPP/cellulose composite filaments by single-screw extrusion at 180 °C, where the cellulose sources were wastepaper, cardboard, and wood flour. The tensile strength and elastic modulus of 3D-printed composites were obtained in the range of 13–18 MPa and 1100–1500 MPa, respectively, where a 10 wt% cellulose composite filament showed about 38% increase in elastic modulus over neat PP.

Conversely, Stoof et al. [43] fabricated rPP/harakeke fiber and rPP/hemp composite filaments by twin-screw extrusion at 180 °C, where the harakeke and hemp fibers were obtained by alkali digestion. A 30 wt% harakeke composite 3D-printed structure exhibited a tensile strength and Young’s modulus of 39 MPa and 2.8 GPa, respectively, which is about a 74% and 214% increase from neat PP. Lately, other composite systems, such as rPP/cacao bean shell (CBS) particles [44] and rPP/rice husk (RH) [45], have also been investigated. CBS addition reduced the characteristic warping effect in 3D printing of rPP by 67% and improved the tensile strength and fracture strain of rPP specimens printed at 90° (compared to 0°), where higher particle fracture, filler–matrix debonding, and matrix breakage were observed for samples printed at 0° (Figure 4b) [44]. Conversely, rPP/RH composite that was 3D printed at 0° exhibited a relatively higher tensile strength compared to the 90° 3D-printed sample [45]. In a separate study, Zander et al. [46] processed blends of waste PP, PET, and PS into filaments for 3D printing, and studied the effect of styrene ethylene butylene styrene (SEBS) and maleic anhydride-functionalized SEBS as the compatibilizer on the resulting mechanical and thermal properties. The 3D-printed rPP/PET and rPP/PS blends exhibited the highest tensile strength of 24 MPa and 22 MPa, respectively, which is about 26% and 16% increase from neat PP. Recently, post-processing heat treatment of 3D-printed parts has also been shown to enhance the mechanical properties [49]. Moreover, the 3D-printed products can also be reprocessed for nanocomposite formulation after the desired use.

### 2.4. Catalytic Transformation of Waste Plastic for the Production of Fine Chemicals and Carbon Materials

Plastic waste can be used as an important feedstock material for the preparation of value-added platform chemicals. Conventional approaches used for chemical recycling of plastic waste include pyrolysis (typically using inert atmosphere at 400–800 °C), gasification (typically using air, oxygen, or steam at >700 °C), and solvolysis (typically using solvent medium at 80–280 °C) [50]. However, these techniques are energy intensive and face several challenges, including higher temperature, lower control over product selectivity, longer duration, etc. To overcome such difficulties, researchers have explored the application of different catalysts for transformation of plastic waste into various value-added products under milder conditions (i.e., upcycling) [51]. A summary of catalysts applied for plastic waste upcycling is given in Table 3.

#### 2.4.1. Nanocatalyzed Pyrolysis

Raw materials can be produced via catalytic pyrolysis of plastic waste, which is performed in presence of a Lewis acid catalyst or solid acid catalyst, where mostly aromatic products or gases are produced by catalyzed carbocation, isomerization, aromatization, or crackling [51]. For example, when a zeolite (HZSM-5) catalyst is used in HDPE waste pyrolysis, the reaction occurs through carbocation formation, which enhances the formation of shorter hydrocarbon chains, thereby increasing the gas (ethylene) yield from 15 wt% (conventional) to 77 wt% (catalyzed) at 600 °C [52]. The use of a catalyst not only increase the reaction rate and product selectivity, but also decreases the production of harmful chemicals during pyrolysis. Carbon-supported Platinum (Pt) NPs’ catalyst can effectively reduce (by decyclization and/or free-radical mechanism) the concentrations of toxic biphenyl derivatives and polycyclic hydrocarbons produced during the pyrolysis of PET waste by 56% and 107%, respectively [87]. Nanocatalyzed pyrolysis carbonization of plastic waste to produce carbonaceous nanomaterials and hydrogen (H_2_) can be potentially achieved using transition 3d-metals (e.g., Fe, Co, Ni, and Mo) containing nanocatalysts. Cai et al. [53] performed pyrolysis catalysis of polyolefin (PE and PP) and PS using an aluminum oxide-supported iron (Fe/Al_2_O_3_) nanocatalyst, and they reported that more nanotubes and pure carbon nanotubes (CNTs) can be produced from polyolefin, whereas more amorphous carbon and H_2_ can be produced from PS. However, coke deposition and metal sintering at higher temperatures affect the stability of the catalyst, particularly for Ni, which can be resolved by using bimetallic catalysts [88]. Consequently, the composition of the bimetallic nanocatalyst greatly influences the structure, electronic state, and coordination environment, which increases the quality of carbon formed and the co-production of H_2_ [51]. For instance, Awadallah and co-workers [54] demonstrated the production of a mixture of large-diameter multiwalled CNTs (MWCNTs) and carbon nanofibers (CNFs) by pyrolysis catalysis of PP waste using a La_2_O_3_-supported Ni-Cu bimetallic nanocatalyst with a Ni:Cu ratio of 4:1. The group also reported the production of nanostructured carbon materials from LDPE waste using magnesium oxide-supported iron–molybdenum (Fe-Mo/MgO) bimetallic nanocatalysts. An intermediate loading of Fe and Mo produced mainly CNTs, accompanied with CNFs and graphene nanosheets (GNSs) as hybrid materials, whereas a higher Fe or Mo loading promoted the formation of both CNTs and CNFs [55]. Furthermore, the size and structure of the catalyst support is also reported to influence the quality of CNTs produced from plastic waste [89].

#### 2.4.2. Nanocatalyzed Gasification

Catalytic gasification is another route to upcycling of plastic waste, which can be performed in the presence of a solid acid-supported metal catalyst, where combustible gases with high calorific value, such as H_2_, methane (CH_4_), and carbon monoxide (CO), are produced by catalyzed pyrolysis, crackling, and reforming by oxygen [51]. The Ni-based nanocatalyst is the most effective one for gasification/reform of plastic waste into H_2_-rich syngas as it exhibits high ability for the coordination and activation of C–H and C–C bonds, which suppresses coke deposition [90]. Asadi et al. [56] investigated the production of syngas by gasification of plastic waste mixture (PET, PE, and PP) using the Ni/zeolite (ZSM-5) nanocatalyst under a different ratio of nitrogen/oxygen, and in the presence of a second promotor. An increase in oxygen ratio (10%) and the use of lanthanum as a second promotor increased syngas production up to 130.7 mmol/g_plastic_ at 850 °C. Similarly, catalytic steam reforming of HDPE waste using the Ni/ZSM-5 nanocatalyst has been reported to produced 100.72 mmol/g_plastic_ of syngas at 850 °C [57]. On the other hand, Wu et al. [58] designed and tested a catalyst comprising a Ni core and CeO_2_–ZrO_2_ shell for H_2_-rich syngas (66.81 mol% of H_2_ at 800 °C) production from HDPE waste by gasification. In a separate study, Zhang et al. [59] synthesized a carbon nanofiber (CNF)/porous carbon (PC)-supported bimetallic (Ni-Fe) catalyst (Ni-Fe/CNT-PC) and demonstrated its potential for production of syngas 63.17 mmol/g_plastic_) by gasification of HDPE waste. The gas products largely consist of H_2_ (33.66 mmol/g_plastic_); a considerable amount of CO, CH_4_, and CO_2_; and trace amounts of the C_2+_ component. Lately, the Al_2_O_3_-supported Ni nanocatalyst has also been reported to enhance the production of syngas by gasification of polyolefin waste. For instance, Arregi et al. [60] investigated the efficiency of the Ni/Al_2_O_3_ nanocatalyst for steam reforming/gasification of PP waste and reported a space time of 16.7 g_catalyst_ min g_plastic_^–1^ was required for full conversion and high H_2_ production. Moreover, when metal oxides, such as lanthanum oxide (La_2_O_3_) and cerium oxide (CeO_2_), were used as the second promotor, a significant increase in H_2_ production was observed. In addition, microwave, plasma, and supercritical water have also been reported to enhance nanocatalyzed gasification [91].

Unlike nanocatalyzed pyrolysis and gasification, which produce value-added products like CNTs and H_2_-rich syngas, solvolysis produces monomers by depolymerization of plastic waste in the presence of solvent, which is largely used as feedstock for polymer resynthesis. Moreover, solvolysis is largely targeted for depolymerization of hydrolysable plastics like PET and PU, and it faces challenges like longer completion time, smaller batch size, and difficulty in separating feedstock impurities including the catalyst used. Therefore, solvolysis is commonly considered for plastic waste recycling and combined with other techniques for upcycling [51].

#### 2.4.3. Nanocatalyzed Hydrogenolysis and Hydrocracking

Catalytic hydrogenolysis of plastic waste has been performed in the presence of a solid acid-supported metal catalyst, where the C–C or C–hetero atom bond of plastic is cleaved by H_2_ to produce value-added hydrocarbons like alkanes, cycloalkanes, alkenes, alkynes, and aromatics [92]. It is also the most extensively studied plastic waste upcycling process in recent years [51]. Pt and ruthenium (Ru) are the two metal catalysts extensively studied for hydrogenolysis of polyolefins. For example, Utami et al. [61] prepared the sulfated zirconium oxide (ZrO_2_)-supported Pt catalyst (Pt/ZrO_2_) and studied the effect of different Pt loading on hydrogenolysis of LDPE plastic waste into liquid fuels, where an increase in activity and selectivity for catalytic hydrocracking was observed with the increase in Pt loading. Jumah et al. [62] demonstrated fast hydrogenolysis (<15 min in a batch system; at 330 °C under 20 bar H_2_) of LDPE waste for high yield (>95% conversion) production of C_4_−C_6_ alkanes using a Pt-impregnated USY zeolite catalyst (Pt/USY). In a separate study, the Pt/SrTiO_3_ catalyst fabricated by depositing (by atomic layer deposition) Pt-NPs on hydrothermally synthesized strontium titanate (SrTiO_3_) perovskite nanocuboids hydrogenolyzed (at 300 °C under 170 psi H_2_) PE waste to lubricants and waxes, characterized by a narrow distribution of oligomeric chains [63]. Conversely, inspired by enzyme-catalyzed conversions of biomacromolecules, Tennakoon et al. [64] fabricated a mesoporous silicon dioxide (mSiO_2_)-based ordered Pt/SiO_2_/mSiO_2_ catalyst (core catalyst/active site/mesoporous shell architecture) and demonstrated its potential for hydrogenolysis of HDPE into a narrow distribution of diesel and lubricant-range alkanes. A single pot catalyst strategy to branched products via adhesive isomerization and hydrocracking of PE (at 250 °C under 30 bar H_2_) was also realized. The method used a Pt-impregnated tungstated zirconia (WZrO_2_) catalyst (Pt/WZrO_2_) to produce alkanes, where an increase in the metal-to-acid site molar ratio shifted the extractable product selectivity (C_1_–C_35_) to heavier hydrocarbons and enhanced branching in the residual polymer [65]. Lately, hydrocracking of polyolefins to produce branched, liquid fuels including diesel, jet, and gasoline-range hydrocarbons using a tungstate–zirconia-supported Pt NPs catalyst (Pt/WO_3_/ZrO_2_) along with HY zeolite catalyst was reported by Liu et al. [66]. The 2 h reaction (at temperatures as low as 225 °C under 30 bar H_2_) with a yield of up to 85% involved tandem catalysis with initial activation of the polyolefins over Pt, followed by polymer cracking over the acid sites of WO_3_/ZrO_2_ and HY zeolite, isomerization of hydrocarbon over WO_3_/ZrO_2_ sites, and subsequent hydrogenation of olefin intermediates over Pt.

Ruthenium (Ru) NPs supported on a carbon (Ru/C) catalyst (at 5 wt%) have been realized for hydrogenolysis of Csp^3^–Csp^3^ bonds of polyolefin waste into short-chain hydrocarbons (alkanes, liquid fuels, and lubricants with yield in the range of 45–68 wt%) at a wide range of experimental conditions (in the range of 200–280 °C, 20–60 bar H_2_, and 1–16 h) [67,68,69]. Improvements in yield of up to 90% have also been achieved for hydrogenolysis of PE and PP waste, using different metal oxide supports for Ru, such as CeO_2_ [70], TiO_2_ [71], WZrO_2_ [72] and niobium pentoxide (Nb_2_O_5_) [73]. The Ru/ZrO_2_ catalyst was reported to be more effective (3-fold higher activity) than Ru/CeO_2_ catalyst for hydrogenolysis of LDPE [74]. Recently, Lee et al. [75] reported hydrocracking (at 350 °C using near-stoichiometric amounts of H_2_) of PE, PP, and PS waste to grid-compatible methane with high selectivity and purity (>97%) using a Ru-modified zeolite (Ru/FAU) catalyst. Moreover, density functional theory (DFT) revealed a chain-end initiation process with a Ru-dominated reaction pathway. On the other hand, Hongkailers et al. [76] demonstrated a one-pot reaction combining PET waste depolymerization and hydrodeoxygenation (at 340 °C under 30 bar initial H_2_ pressure) via C–O cleavage to produce arenes (xylene and toluene) using a Co/TiO_2_ catalyst. CuNa/SiO_2_ has been demonstrated to be able to convert PET to para-xylene (with high yield) and ethylene glycol by using methanol as both the solvent and H-donor resource [77]. Lately, high yield (85%) production of arenes by hydrogenolysis of PET and PS waste over a Ru/Nb_2_O_5_ catalyst has also been reported [73]. Upcycling of PET waste into cycloalkanes and aromatics via an integrated tandem (comprises at least two consecutive reactions) solvolysis–hydrogeneration process has also been reported [78]. In this process, the PET waste is first hydrolyzed to dimethyl terephthalate (DMT) in the absence of any catalyst. Next, the DMT is liquefied to dimethyl cyclohexane-1,4-dicarboxylate by solvent-free hydrogenation (over Pt/C catalyst) and subsequently hydrodeoxygenated (over Ru-Cu/SiO_2_ catalyst) to valuable gasoline and jet fuel range C_7_–C_8_ cycloalkanes and aromatics. Recently, Zhang et al. [79] demonstrated PE waste upcycling to valuable long-chain alkyl aromatics by a tandem hydrogenolysis–aromatization process. In this process, the PE waste is first combined with the catalyst Pt/γ-Al_2_O_3_ in an autoclave (without no added solvent or H_2_) and heated to 280 °C for 24 h (Figure 5a), where aromatization reaction of short hydrocarbons generates in situ H_2_, which is compatible with the hydrogenolysis of long hydrocarbon chains under a reductive environment. Next, the obtained liquid/wax product (80% by mass) was dissolved in hot chloroform to obtain the long hydrocarbons. In addition, the application of molybdenum (Mo)-based catalyst for the production of terephthalic acid (TPA) and ethylene by hydrogenolysis (at 260 °C under atmospheric H_2_) of PET waste has also been reported [93,94].

#### 2.4.4. Nanocatalyzed Photoreforming

Another promising method for plastic waste upcycling is photoreforming, where depolymerization of plastic is performed under atmospheric conditions using light and a photoactive catalyst. UV-absorbing semiconductor nanoparticles are a promising photocatalyst for the depolymerization of PE and PS waste. For instance, Nabi et al. [80] studied the efficiency of photocatalytic titanium oxide (TiO_2_) NP film for the depolymerization and complete mineralization of PP waste. While CO_2_ was found as the main end-product of PE photodegradation, the generation of hydroxyl, carbonyl, and carbon–hydrogen groups was obtained from PS photodegradation. Conversely, Tofa et al. [81] demonstrated depolymerization of LDPE waste using photocatalytic zinc oxide (ZnO_2_) nanorods, which produced hydroperoxides, peroxides, carbonyl, and unsaturated groups. In a separate study, Reisner and co-workers [82] produced H_2_ by the photoreforming of ground PET powder using solar light-simulated cadmium selenide/cadmium oxide (CdS/CdO_x_) quantum dots. The PET was first hydrolyzed in potassium hydroxide (KOH) solution to produce terephthalate, ethylene glycol (EG), and isophthalate, where EG was then photoreduced to yield H_2_, and the rest was photo-oxidized to produce formate, glycolate, ethanol, acetate, and lactate. The group also utilized a carbon nitride/nickel phosphide (CNx/Ni_2_P) catalyst for visible-light-driven PET reforming to produce clean H_2_ (Figure 5b), where PET acts as an electron donor and is oxidized by the excited photocatalyst (CNx) to other organic molecules. The photogenerated electrons are then transferred from the CNx co-catalyst (Ni_2_P) and reduce water to H_2_ [83]. Lately, upcycling of PE waste into gaseous hydrocarbons via an integrated tandem solvolysis–photoreforming process has also been realized [84]. In this process, the PE waste is first oxidized to dicarboxylic acids (largely succinic and glutaric acid) using nitric acid, followed by a photocatalysis step. Photocatalysis of succinic and glutaric acid was performed using a UV light-absorbing Pt/TiO_2_ or platinized carbon nitride photocatalyst to obtain ethane/ethylene (with propanoic acid intermediate) and propane/propylene (with butyric acid intermediate), respectively.

#### 2.4.5. Nanocatalyzed Electroreforming

Electroreforming is a relatively less explored method for plastic waste upcycling, where depolymerization of plastic involves oxidative cleavage of the carbon–carbon bond at the anode and H_2_ evolution at the cathode [96]. Shi et al. [85] reported successful electrocatalytic reforming of PET waste into value-added chemical products (terephthalate, carbonate, and H_2_) using a Pd-modified Ni foam (Pd/NF) catalyst (Figure 5c). The process involved pre-treatment of PET in KOH to release terephthalate and EG, followed by electrocatalytic oxidation of EG to H_2_ and carbonate (95% selectivity). Similarly, Zhou et al. [86] demonstrated electrocatalytic reforming of PET waste into TPA, potassium diformate (selectivity >80%), and H_2_ using a CoNi_0.25_P catalyst.

Among the various catalytic polymer processing methods developed, only catalytic pyrolysis, gasification, and crackling have reached a technology readiness level of 8–9 (fully developed and tested, and in commercial operation).

### 2.5. Industrial Biotechnology

Recent developments in biotechnology, such as enzyme-mediated biocatalytic depolymerization of plastic waste into value-added chemicals and utilization of depolymerized plastic waste as feedstock material for microbe-mediated biopolymer synthesis have emerged as sustainable and efficient methods for plastic waste upcycling [97,98]. The “green” nature of these transformations without the use of any hazardous substances provides an eco-friendly route for plastic waste upcycling.

#### 2.5.1. Enzymatic Depolymerization

To date, various plastic-degrading enzymes have been discovered from different microbial sources (Figure 6a) and have been extensively studied and engineered for degradation of both hydrolysable (e.g., PET, and PU) and non-hydrolysable (e.g., PE, PP, and PS) plastics. Synthetic plastic with a relatively higher crystallinity is more resistant to enzymatic attack compared to biogenic polymers. Therefore, protein engineering has been increasingly utilized to design, engineer and synthesize plastic-degrading enzymes with better catalytic efficiency [97]. The class of enzymes “hydrolases”, which are used for degradation of hydrolysable plastics, include cutinases, lipases, carboxylesterases, esterases, and proteases, which attack the hydrolysable bonds of plastics like esters or amides for depolymerization [99]. The general reaction of a hydrolase enzyme for breakdown of a product (X-Y) in the presence of water is given as [100]:X-Y + H_2_O → X-OH + Y-H

The depolymerization of plastic waste by enzymes is a two-step process, where the enzyme first adheres to plastic surface by hydrophobic interaction, followed by the hydrolytic cleavage of the long polymer chains of plastic by the active site of the enzyme into smaller monomers or dimers, which can be accumulated or consumed by the microbial organism as a carbon source [101]. On the other hand, non-hydrolysable plastics with an inert C-C backbone are highly resistant to biological cleavage and can only be broken down via high-energy oxidation reactions. Therefore, enzymatic degradation of non-hydrolysable plastics is very limited, and often, catalytic-, photo-, and thermal-degradation mediated by radical mechanisms are performed prior to enzymatic degradation [99]. While the enzymatic attack on hydrolysable plastics is generally endo-type (random internal scission), the enzymatic attack on non-hydrolysable plastics is exo-type (end of chain scission) [99].

Yoshida et al. [103] first reported the unusual ability of a newly isolated bacterium, *Ideonella sakaiensis* 201-F6, which could degrade PET and assimilate its monomers using two enzymes, designated as PETase and MHETase. This later led to the development of thermo-stable PETase from *I. sakaiensis* by rational protein engineering for highly efficient PET degradation [104], where PET has a thermal glass transition temperature in the range of 65–80 °C, at which wild-type enzymes are generally unstable. Genetically engineered microorganisms, including *Psedomonas aestusnigri*, *Thermobifida fusa*, and *Clostridium botulinum,* have also been developed to produce hydrolases, such as PETase, cutinase and esterase, for improved depolymerization efficiency [97]. The enzymatic degradation of PET generates monomers TPA and EG, which can be further used to produce PET, be converted into value-added chemicals or can be potentially used as feedstock for synthesis of biodegradable plastics. For example, Tournier et al. [105] demonstrated PET depolymerization (90% efficiency) using an engineered PET hydrolase, and the produced monomer TPA was then used to synthesis PET, which was ultimately blown into bottles with better lightness values (87.5%) than those of the minimal standard for PET bottles (>85%). Conversely, Sadler and Wallace [106] demonstrated direct upcycling of PET-derived TPA into value-added vanillin (with 79% conversion efficiency) in aqueous media (pH 5.5–7) at room temperature using engineered *E. coli*. Kim and co-workers [102] reported the conversion of five different higher-value aromatics and aromatic-derived compounds (gallic acid (GA), pyrogallol, catechol, muconic acid (MA), and vanillic acid (VA), with yields in the range of 32.7–92.5% from PET waste-derived TPA using engineered *Escherichia coli*, whereas waste PET-derived EG was fermented to glycolic acid (GLA) using *Gluconobacter oxydans*, as shown in Figure 6b. In a separate study, the authors also demonstrated depolymerization of PET waste to produce TPA (31.0 g/L, 62.8%, mol/mol) and EG (11.7 g/L, 63.3%, mol/mol) via a one-pot chemo-bioprocess integrating chemical glycolysis (using a biocompatible catalyst, Betaine), enzymatic hydrolysis, and bioconversion of TPA and EG to protocatechuic acid (PCA) and GLA, respectively [107]. Recently, Kang et al. [108] reported the synthesis of 2-pyrone-4,6-dicarboxylic acid (PDC) (a valuable monomer for biodegradable plastics) from PET waste-derived TPA via a comprehensive chemo-microbial hybrid process using two recombinant *E. coli* strains. On the other hand, Magnin et al. [109] studied the poly(ester urethanes) and poly(ether urethanes) depolymerization efficiency of a collection of 50 hydrolases and their mixtures, and they reported that esterase (E3576) and amidase (E4143) enzymes were able to effectively hydrolyze a waterborne polyester PU dispersion and the urethane bond of a low molar mass molecule, respectively. The highest degradation (33% weight loss after 51 days) was reported for a polycaprolactone polyol-based PU using the enzyme esterase, with 6-hydroxycaproic acid and 4,4′-methylene dianiline recovered as hydrolysis products [109].

#### 2.5.2. Biopolymer Synthesis

Polyhydroxyalkanoates (PHAs) are biodegradable polyesters synthesized by various microorganisms in the presence of an excess amount of carbon sources and lack of macro elements such as phosphorus, nitrogen, trace elements, or oxygen [110]. Several studies have reported the utilization of plastic waste-derived monomers as the carbon source for the bacterial synthesis of PHAs. For example, Kenny et al. [111] reported the synthesis of PHAs using three bacterial strains, namely *P. putida* GO16, *P. putida* GO19, and *P. frederiksbergensis* GO23, utilizing TPA (obtained from pyrolysis of PET waste) as the sole source of carbon and energy. The strains GO16 and GO19 accumulated PHA (predominantly of a 3-hydroxydecanoic acid monomer) at a maximal rate of approximately 8.4 mg PHA/l/h for 12 h, whereas the strain GO23 accumulated PHA (predominantly of a 3-hydroxydecanoic acid monomer) at a lower maximal rate of 4.4 mg PHA/l/h [111]. Recently, Tiso et al. [112] demonstrated the synthesis of PHA (0.014 g_PHA_/g_substrate_) and hydroxyalkanoyloxy-alkanoates (HAAs) using *P. umsongensis* GO16 KS3 utilizing EG and TA (obtained by cutinase depolymerization of PET) as the carbon source. The authors also reported chemo-catalytic synthesis of bio-based poly(amide urethane) (bio-PU) using the obtained HAAs as monomers [113]. In a separate study, Franden et al. [113] synthesized medium-chain-length PHAs (32% yield) using the *P. putida* KT2440 strain and utilizing EG as the carbon source. Conversely, Johnston et al. [114] reported the synthesis of PHA (42% yield) using the bacterial strain *Cupriavidus necator* H16 and utilizing oxidized PP waste as an additional carbon source.

## 3. Conclusions and Outlook

In summary, separate or mixed plastic waste can be effectively converted into value-added products by different upcycling approaches; including vitrimerization, nanocomposite fabrication, 3D printing and chemical and microbial transformations. Vitrimerization, nanocomposite fabrication, and 3D printing technologies can be potentially applied for the fabrication of new products with desired structural, mechanical, and functional properties. On the other hand, chemical and microbial transformation technologies can be adapted to obtain a diverse range of products including carbon allotropes, syngas, H_2_, monomers, liquid fuels, grid-compatible gas streams, lubricants, waxes, and biodegradable polymers. Vitrimers, with their excellent re-processability and responsiveness, can be considered as next-generation smart circular materials, which can be synthesized with affordable technologies using conventional instruments that are operated in the polymer processing industries. Nanocomposite formulations comprising biomass represent a promising approach for the development of sustainable and compostable products, which could be easily implemented or adapted by polymer processing industries with the current infrastructure. However, the adaptability of 3D printing for large-scale production is still a long way away due to the slow build up speed. Emerging technologies like microwave, plasma, and supercritical water could provide synergistic effects to address the limitations for the further development of current technologies like pyrolysis, gasification, and solvolysis. Although the application of nanocatalysts has largely reduced the energy required for the chemical transformation of plastic waste, technological challenges related to the cost and performance of the catalyst, such as reaction and/or product selectivity, durability and/or recyclability, and removal and/or reusability, need to be addressed. Moreover, enzyme-meditated catalysis for plastic degradation and the production of bioplastic from non-conventional feed stocks using microorganisms will significantly reduce the carbon footprint. The envisaged circular economy necessitates the utilization of waste by different methods to minimize waste and ultimately carbon dioxide formation. Therefore, with the goal of ‘sustainable development with closed loop waste management’, the upcycling of plastic waste to biodegradable plastic needs to be part of the new thinking and should be considered as a means of valorization of post-consumer plastic.

## Figures and Tables

**Figure 1 polymers-14-04788-f001:**
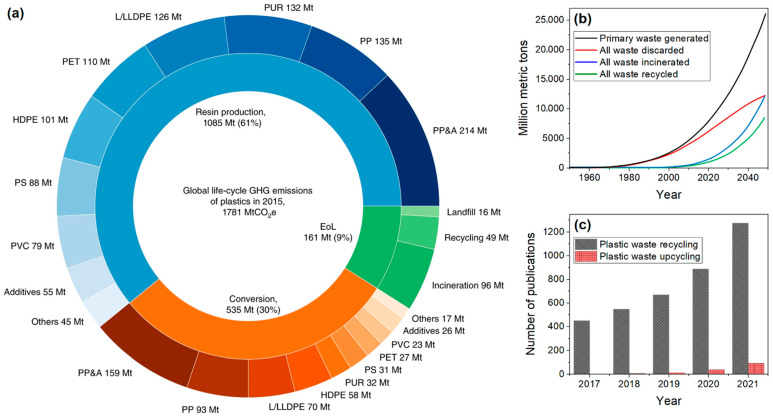
(**a**) Global GHG emissions of conventional plastics by life cycle stage and plastic type. Reproduced with permission from [3]. Copyright 2019, Springer-Nature. (**b**) Estimation of cumulative plastic waste generation and disposal. Adapted from [4], which is licensed under CC BY 4.0. (**c**) Publication trends over the last five years obtained from Web of Science using keywords “Plastic waste recycling” and “Plastic waste upcycling”.

**Figure 2 polymers-14-04788-f002:**
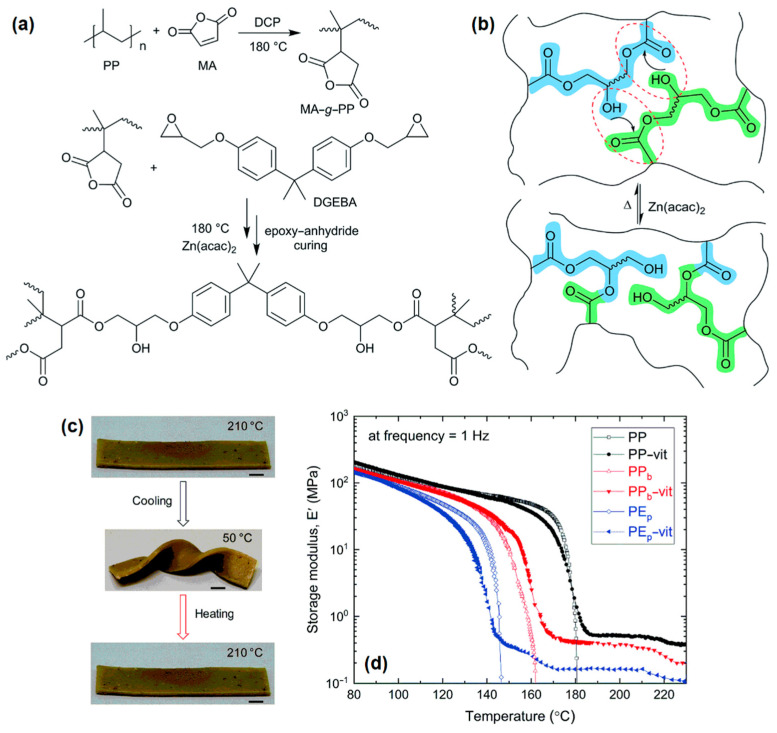
(**a**) Schematic of the vitrimer synthesis using PP bottle waste (PP_b_) through reactive extrusion. The PP_b_ is first grafted with MA using DCP as free radical initiator, followed by crosslinking of MA-grafted rPP with a di-functional epoxy (DGEBA) using Zn(acac)_2_ as catalyst. (**b**) Thermo-reversible associative bond exchange in the rPP vitrimer through transesterification. (**c**) Picture showing shape-memory effect in the rPP vitrimer with response to temperature. Scale bar is 5.0 mm. (**d**) Thermomechanical properties of the PP_b_ and PE packaging waste (PE_p_) vitrimers compared to the original materials. Reproduced with permission from [20]. Copyright 2020, Royal Society of Chemistry.

**Figure 3 polymers-14-04788-f003:**
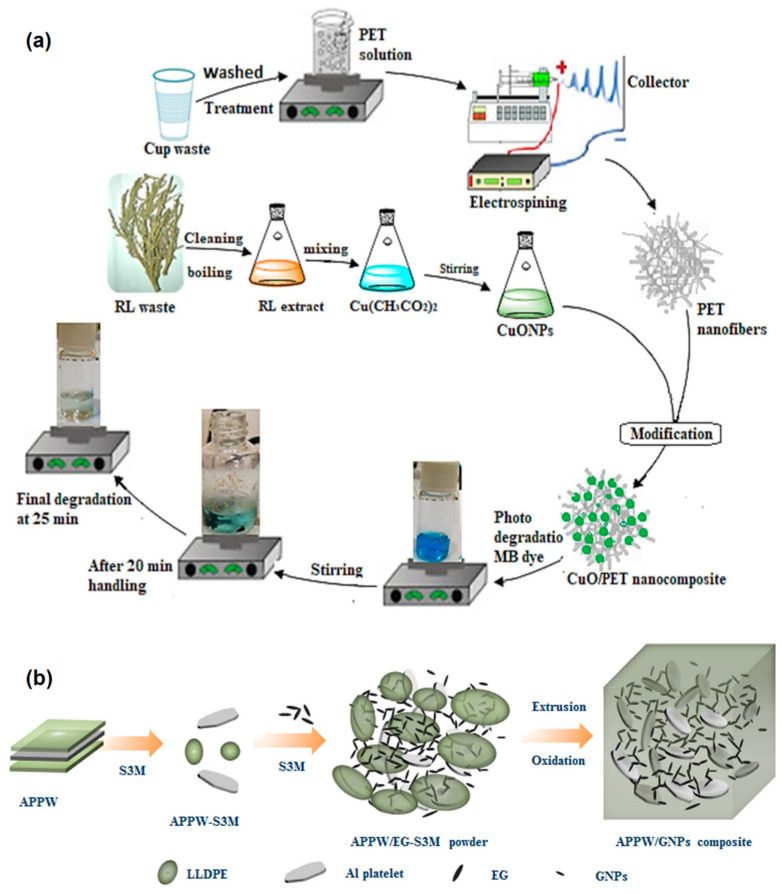
(**a**) Steps of rPET/CuO nanocomposite preparation and methylene blue dye degradation process. Reproduced from [23], which is licensed under CC BY 4.0. (**b**) Schematic of the preparation process of the rLDPE/PET/Graphite NPs composite. Reproduced with permission from [26]. Copyright 2018, American Chemical Society.

**Figure 4 polymers-14-04788-f004:**
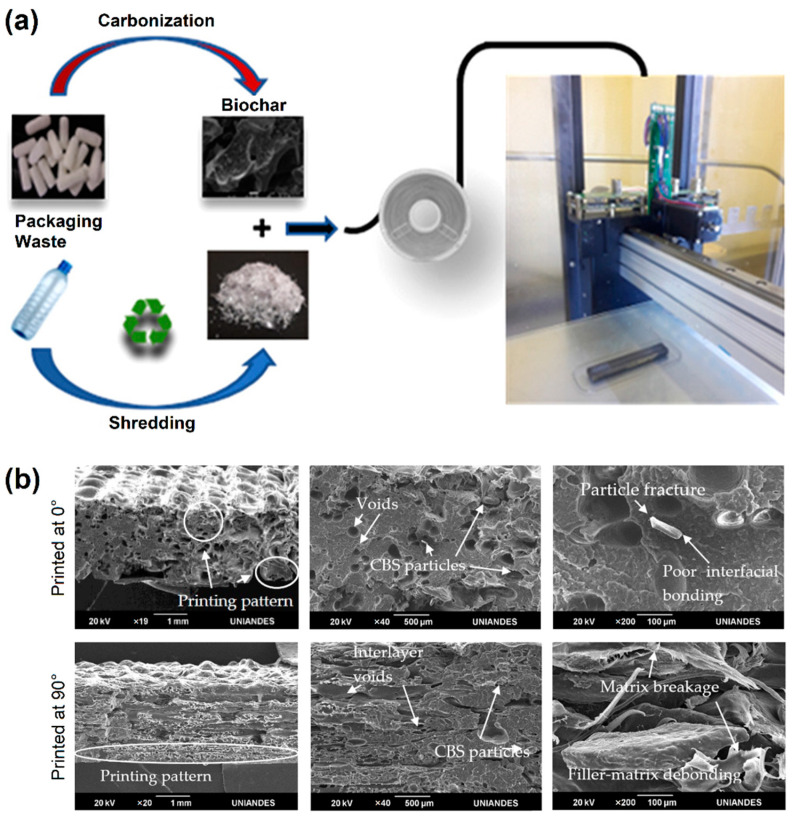
(**a**) Schematic of the preparation and 3D printing process of the rPET/Biochar composite. Reproduced with permission from [37]. Copyright 2018, American Chemical Society. (**b**) Scanning electron microscope images of 3D-printed and tensile-fractured rPP/CBS composite specimens. Reproduced from [44], which is licensed under CC BY 4.0.

**Figure 5 polymers-14-04788-f005:**
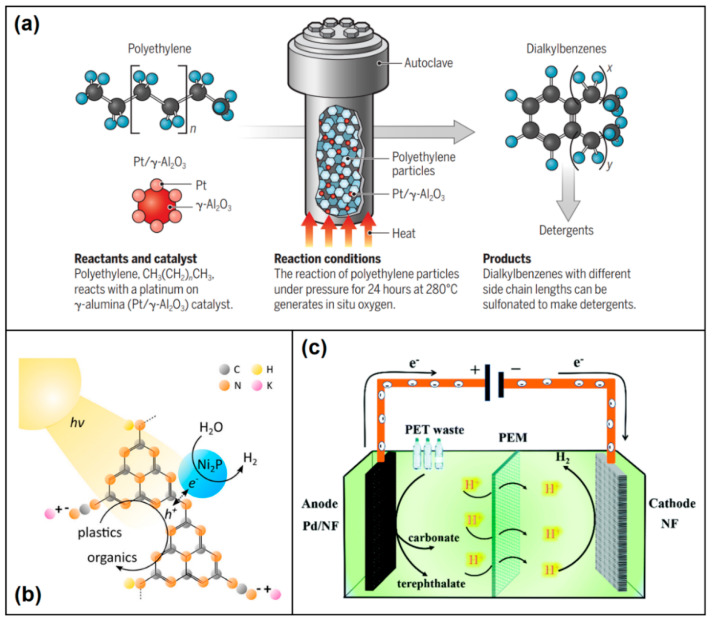
(**a**) Upcycling of PE waste to valuable products, such as detergents, via tandem hydrogenolysis/aromatization using Pt/γ-Al_2_O_3_ catalyst. Reproduced with permission from [95]. Copyright 2020, American Association for the Advancement of Science. (**b**) Schematic of the plastic waste photoreforming process using the CNx/Ni_2_P photocatalyst. Reproduced with permission from [83]. Copyright 2019, American Chemical Society. (**c**) Schematic of the electroreforming of PET into high value-added chemicals and H_2_ fuel. Reproduced with permission from [85]. Copyright 2021, Royal Society of Chemistry.

**Figure 6 polymers-14-04788-f006:**
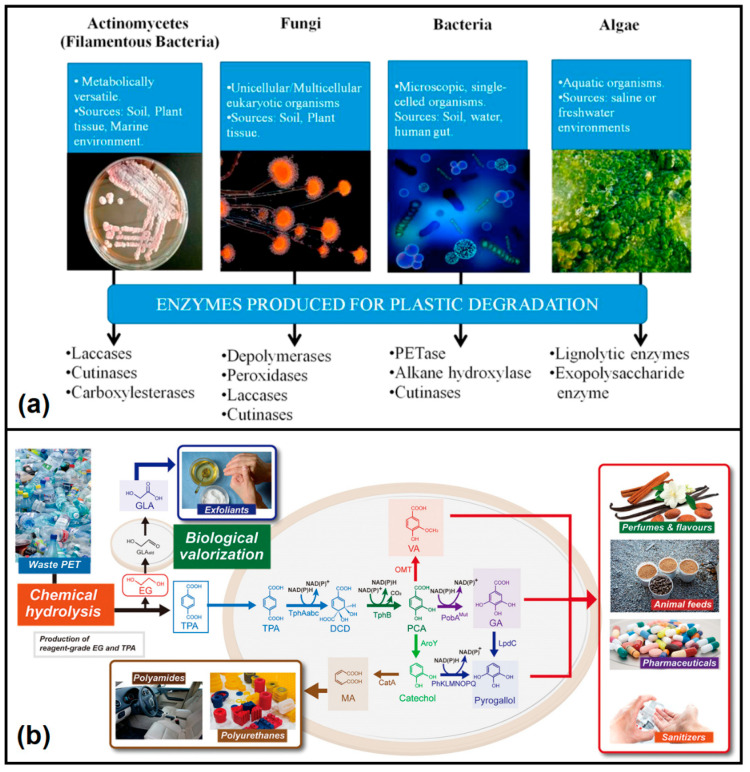
(**a**) Critical enzymes for plastic degradation from different microorganisms. Reproduced with permission from [100]. Copyright 2021, Elsevier. (**b**) Overall scheme for the waste PET biorefinery for upcycling PET. Reproduced with permission from [102]. Copyright 2019, American Chemical Society.

**Table 1 polymers-14-04788-t001:** Value-added composite materials made from plastic waste.

Composites	Processing Method	Obtained Properties	Applications	References
rPET/CuO NPs	Electrospinning and chemical precipitation techniques.	Photocatalytic activity efficiency for removing the methylene blue dye up to 99%.	Water treatment and filtration	[23]
rHDPE/CuO NPs	Melt mixing and compression molding.	Increased electron density, mass attenuation coefficient, and effective atomic number for γ-ray energies with point sources 356 keV from ^133^Ba, 662 keV from ^137^Cs, and 1332 keV from ^60^Co.	Radioactive source shielding	[24]
rLDPE/SiO_2_/TiO_2_ NPs	Melt extrusion, granulation, and compression molding.	Tensile strength of 8.4 MPa, and UV protection factor of 1500+.	UV shielding	[25]
rLDPE/PET/Al/ Graphite NPs	Shear milling, melt extrusion, and injection molding.	Thermal conductivity of 1.7 W/mK, and electrical conductivity of 10^−10^ S/cm.	Electronic packaging	[26]
rLDPE/PA/Al nanoflake	Powder mixing and compression molding.	Thermal conductivity in the range of 1.4–4.8 W/mK, and electrical conductivity of 10^−13^ S/cm.	Electronic packaging	[27]
rPS/SnO_2_ NPs	Thermally induced phase separation.	Photodegradation efficiency of rhodamine B dye under UV irradiation up to 98.2%.	Water treatment and filtration	[28]
rPS/TiO_2_ NPs/Al microparticles	Solution mixing and electrospinning.	Water contact angle of 157° (superhydrophobic), and daily water productivity of >1.35 L/m^2^.	Fog water-harvesting	[29]

**Table 2 polymers-14-04788-t002:** Summary of value-added structures that are 3D printed by FDM using composite or blend filaments made from plastic waste.

Filament Material	Extrusion Conditions	3D Printing Parameters	Mechanical Properties of Printed Structures	References
rPET/biochar composite	Single screw extrusion at 250 °C.	Bed temperature of 50 °C, nozzle temperature of 270 °C, layer height of 0.4 mm, print speed of 50 mm/s, nozzle diameter of 0.6 mm.	Tensile strengths in the range of 46–52 MPa, elastic modulus in the range of 0.7–0.9 GPa.	[37]
rPET/cellulose fiber composite	Twin screw extrusion with screw speed of 38–43 rpm, feed port at 200 °C, adjacent zone at 260 °C, main zones at 240 °C, die at 220 °C.	Nozzle temperature of 260 °C, print speed of 30 mm/s.	Impact resistance of 23.30 J/m, and impact strength of 2268 J/m^2^.	[38]
rPET/CCFs composite	Co-extrusion.	Bed temperature of 80 °C, nozzle temperature of 230 °C, layer height of 0.2 mm, print speed of 300 mm/s, nozzle diameter of 0.4 mm.	Tensile strength of 604.5 MPa, flexural strength of 318.6 MPa.	[39]
rHDPE/PP/PP-MAh blend	Single screw extrusion with screw speed of 20 rpm, feed port at 140 °C, adjacent zone at 150 °C, main zones at 160 °C, die at 155 °C.	Bed temperature of 105 °C, nozzle temperature of 215 °C.	Tensile yield stress of 4.78 MPa, strain of 38.1%.	[40]
rHDPE/CF composite	Twin screw extrusion with screw speed of 30 rpm, feed port at 180 °C, adjacent zone at 185 °C, main zones at 190 °C, die at 200 °C.	Bed temperature of 80 °C, nozzle temperature of 290 °C, nozzle diameter of 0.8 mm.	Tensile yield stress in the range of 18–21 MPa, tensile strengths in the range of 37–64 GPa.	[41]
rPP/cellulose composite	Twin screw extrusion with screw speed of 100 rpm, feed port at 140 °C, adjacent zone at 170 °C, main zones at 180 °C, die at 175 °C.	Bed temperature of 100 °C, nozzle temperature of 220 °C, layer height of 0.2 mm, print speed of 20–50 mm/s, nozzle diameter of 0.8 mm.	Tensile strengths in the range of 13–18 MPa, elastic modulus in the range of 1100–1500 MPa.	[42]
rPP/harakeke fibers, and rPP/hemp fibers composites	Twin screw extrusion with screw speed of 50 rpm, feed port at 150 °C, adjacent and main zones at 170 °C, die at 180 °C.	Nozzle temperature of 230 °C, print speed of 50 mm/min, nozzle diameter of 1.0 mm.	PP/harakeke fibers exhibited tensile strength and Young’s modulus in the range of 27–39 MPa and 1612–2767 MPa, respectively, whereas PP/hemp fibers were in the range of 28–38 MPa and 1683–2681 MPa.	[43]
rPP/CBS composite	Twin screw extrusion with screw speed in the range of 6–13 rpm, feed port at 175 °C, and the die at 190 °C.	Bed temperature of 90 °C, nozzle temperature of 250 °C, layer height of 0.25 mm, print speed of 60 mm/s, nozzle diameter of 0.8 mm.	Tensile strengths in the range of 8–15 MPa.	[44]
rPP/RH composite	Twin screw extrusion with screw speed of 9 rpm, feed port at 180 °C, adjacent zone at 185 °C, main zones at 190 °C, die at 195 °C.	Bed temperature of 80 °C, nozzle temperature of 240 °C, print speed of 60 mm/s, nozzle diameter of 0.8 mm.	Tensile strengths in the range of 5–14 MPa.	[45]
rPP/rPET, and rPP/rPS blend	Twin screw extrusion with screw speed of 25 rpm, feed port at 140 °C, adjacent zone at 170 °C, main zones at 240 °C, die at 245 °C.	Bed temperature of 100 °C, nozzle temperature of 260 °C, layer height of 0.2 mm, print speed of 20–50 mm/s, nozzle diameter of 0.5 mm.	PP/PET exhibited maximum tensile strength and elastic modulus of 24 MPa and 980 MPa, respectively, whereas PP/PS exhibited 23 MPa and 1459 MPa.	[46]

**Table 3 polymers-14-04788-t003:** Summary of catalysts applied for the valorization of plastic waste.

Plastic	Catalyst	Process	Products	Reference
HDPE	HZSM-5 zeolite	Pyrolysis	Ethylene	[52]
PE, PP, PS	Fe/Al_2_O_3_	Pyrolysis	Amorphous carbon, carbon nanotubes, hydrogen	[53]
PP	Ni-Cu/La_2_O_3_	Pyrolysis	Multiwalled carbon nanotubes, carbon nanofibers	[54]
LDPE	Fe-Mo/MgO	Pyrolysis	Carbon nanotubes, carbon nanofibers, graphene	[55]
PET, PE, PP	Ni/ZSM-5	Gasification	Syngas	[56,57]
HDPE	Ni/CeO_2_–ZrO_2_	Gasification	Hydrogen-rich syngas	[58]
HDPE	Ni-Fe/CNT-PC	Gasification	Hydrogen-rich syngas	[59]
PP	Ni/Al_2_O_3_	Gasification	Hydrogen-rich syngas	[60]
LDPE	Pt/S-ZrO_2_	Hydrogenolysis	Liquid fuels	[61]
LDPE	Pt/USY zeolite	Hydrogenolysis	Alkanes	[62]
PE	Pt/SrTiO_3_	Hydrogenolysis	Lubricants	[63]
HDPE	Pt/SiO_2_/mSiO_2_	Hydrogenolysis	Alkanes	[64]
LDPE	Pt/WZrO_2_	Hydrogenolysis	Alkanes	[65]
LDPE	Pt/WO_3_/ZrO_2_/HY zeolite	Hydrogenolysis	Liquid fuels	[66]
PE, PP	Ru/C	Hydrogenolysis	Alkanes, liquid fuels, lubricants	[67,68,69]
PE, PP	Ru/CeO_2_	Hydrogenolysis	Liquid fuels	[70]
PP	Ru/TiO_2_	Hydrogenolysis	Lubricants	[71]
LDPE	Ru/WZrO_2_	Hydrogenolysis	Alkanes	[72]
PET, PS	Ru/Nb_2_O_5_	Hydrogenolysis	Arenes	[73]
LDPE, HDPE, PP	Ru/ZrO_2_	Hydrogenolysis	Alkanes, liquid fuels	[74]
LDPE, PP, PS	Ru/FAU zeolite	Hydrogenolysis	Grid-compatible gas streams	[75]
PET	Co/TiO_2_	Hydrogenolysis	Arenes	[76]
PET	CuNa/SiO_2_	Hydrogenolysis	Alcohol, aromatics	[77]
PET	Pt/C, Ru-Cu/SiO_2_	Tandem solvolysis–hydrogeneration	Cycloalkanes, aromatics	[78]
PE	Pt/γ-Al_2_O_3_	Tandem hydrogenolysis–aromatization	Long-chain alkylaromatics	[79]
PP, PS	TiO_2_	Photoreforming	Hydroxyl, carbonyl, and carbon-hydrogen groups	[80]
LDPE	ZnO_2_	Photoreforming	Hydroperoxides, peroxides, carbonyl, and unsaturated groups	[81]
PET	CdS/CdO_x_	Photoreforming	Hydrogen	[82]
PET	CNx/Ni_2_P	Photoreforming	Hydrogen	[83]
PE	Pt/TiO_2_	Tandem solvolysis–photoreforming	Alkene, alkane	[84]
PET	Pd/NF	Electroreforming	Hydrogen	[85]
PET	CoNi_0.25_P	Electroreforming	Hydrogen	[86]

## Data Availability

Not applicable.

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
