# Peer review of "Plastic Waste Upcycling: A Sustainable Solution for Waste Management, Product Development, and Circular Economy"

_polymers, 2022, doi:10.3390/polym14224788_

Round 1
Reviewer 1 Report
The review “Plastic Waste Upcycling: a Sustainable Solution for Waste Management, Product Development, and Circular Economy“ is well written, well structured, and with an exhaustive bibliography, but refers only to the last four years.
I think that the review can be strongly improved by writing a chapter with a short “summary” of the state of the art of the upcycling of plastic waste before the last four years.
Author Response
Comment: The review “Plastic Waste Upcycling: A Sustainable Solution for Waste Management, Product Development, and Circular Economy’’ is well written, well structured, and with an exhaustive bibliography, but refers only to the last four years.
I think that the review can be strongly improved by writing a chapter with a short “summary” of the state of the art of the upcycling of plastic waste before the last four years.
Response: As suggested by the reviewer, a short “summary” of the state of the art of the upcycling of plastic waste before the last four years has been added (line 79-84) with an additional reference (no. 13) in the revised manuscript.

Reviewer 2 Report
In this review, the authors summarized recent advancements in plastic waste upcycling, including vitrimerization, nanocomposite fabrication, additive manufacturing, catalytic transformation, and industrial biotechnology, envisaged with technical challenges, future developments, and new circular economy opportunities. This review is well written and structured. I recommend this paper before addressing the following comments.
1. Ru/ZrO2 catalyst was reported to be more effective than Ru/CeO2 catalyst. Pls add it. Appl. Catal. B 318(2022) 121870.
2. CuNa/SiO2 was demonstrated to be able to convert PET and PBT to para-xylene with high yield by using methanol as solvent and H-donor resource. Pls add it in the review. Nature Communications, 2022, 13, 3343.
Author Response
Comment: In this review, the authors summarized recent advancements in plastic waste upcycling, including vitrimerization, nanocomposite fabrication, additive manufacturing, catalytic transformation, and industrial biotechnology, envisaged with technical challenges, future developments, and new circular economy opportunities. This review is well written and structured. I recommend this paper before addressing the following comments.
- Ru/ZrO2 catalyst was reported to be more effective than Ru/CeO2 catalyst. Pls add it. Appl. Catal. B 318(2022) 121870.
- CuNa/SiO2 was demonstrated to be able to convert PET and PBT to para-xylene with high yield by using methanol as solvent and H-donor resource. Pls add it in the review. Nature Communications, 2022, 13, 3343.
Response: The above two comments of the reviewer have been addressed with additional text (line 412-414, and line 421-423) and references (no. 74, and 77) in the revised manuscript.

Reviewer 3 Report
The review "Plastic Waste Upcycling: A Sustainable Solution for Waste Management, Product Development, and Circular Economy" is a well-organized material on the topic of recent developments in the processing of plastic waste, such as vitrimerization, nanocomposite production, additive manufacturing, catalytic transformation and industrial biotechnology, covering technical challenges, future developments and a new circular economy. This article certainly corresponds to Polymers. The text of the article is presented not just correctly, but taking into account modern terminology, with an assessment of the practical value of each type of plastic waste processing. The text is combined with a large number of figures and extensive tables. The list of references exceeds 110 pieces, various sources are cited, without preference of any of the authors or publishers.
The scientific novelty of the presented material can be considered the recognition of the practice of disposable use of synthetic polymers and economically unattractive methods of recycling plastic waste. Such an author's approach will attract the attention of researchers who are able to try to eliminate the created problem.
Questions and recommendations that arose while reading the review:
1. Table 2. Is there a difference in the manufacturing technology of composites when natural polysaccharides and synthetic polymers are added? One sentence is enough in the text.
2. What catalytic polymer processing processes have already been scaled up or put into practice? It can be supplemented with one sentence, thereby revealing the great prospects of the processes described.
3. The most attractive section in the review is 2.5. Industrial biotechnology. Reviewer's recommendations: emphasize the "green" nature of these transformations.
Author Response
Comment: The review "Plastic Waste Upcycling: A Sustainable Solution for Waste Management, Product Development, and Circular Economy" is a well-organized material on the topic of recent developments in the processing of plastic waste, such as vitrimerization, nanocomposite production, additive manufacturing, catalytic transformation and industrial biotechnology, covering technical challenges, future developments and a new circular economy. This article certainly corresponds to Polymers. The text of the article is presented not just correctly, but taking into account modern terminology, with an assessment of the practical value of each type of plastic waste processing. The text is combined with a large number of figures and extensive tables. The list of references exceeds 110 pieces, various sources are cited, without preference of any of the authors or publishers.
The scientific novelty of the presented material can be considered the recognition of the practice of disposable use of synthetic polymers and economically unattractive methods of recycling plastic waste. Such an author's approach will attract the attention of researchers who are able to try to eliminate the created problem.
Questions and recommendations that arose while reading the review:
- Table 2. Is there a difference in the manufacturing technology of composites when natural polysaccharides and synthetic polymers are added? One sentence is enough in the text.
- What catalytic polymer processing processes have already been scaled up or put into practice? It can be supplemented with one sentence, thereby revealing the great prospects of the processes described.
- The most attractive section in the review is 2.5. Industrial biotechnology. Reviewer's recommendations: emphasize the "green" nature of these transformations.
Response: The above three questions and recommendation of the reviewer have been addressed with additional text (line 229-232, 486-488, and line 493-495) in the revised manuscript.

Round 2
Reviewer 1 Report
The answer to my comment is satisfying, but verify the DOI of ref 13 (it is not a paper of the reviewer). Please, check the English of the new statement.